# Treatment Outcomes of Pulpotomy in Primary Teeth with Irreversible Pulpitis: A Systematic Review and Meta-Analysis

**DOI:** 10.3390/children11050574

**Published:** 2024-05-10

**Authors:** Galvin Sim Siang Lin, Yu Jie Chin, Rob Son Choong, Sharifah Wade’ah Wafa Syed Saadun Tarek Wafa, Nabihah Dziaruddin, Fadzlinda Baharin, Ahmad Faisal Ismail

**Affiliations:** 1Department of Restorative Dentistry, Kulliyyah of Dentistry, International Islamic University Malaysia, Kuantan Campus, Kuantan 25200, Pahang, Malaysia; 2Department of Paediatric Dentistry and Orthodontics, Faculty of Dentistry, Universiti Malaya, Kuala Lumpur 50603, Malaysia; yujie1211@gmail.com (Y.J.C.); robsonchoong@gmail.com (R.S.C.); nabihahdziaruddin@um.edu.my (N.D.); 3AVISENA Women’s & Children’s Specialist Hospital, Shah Alam 40000, Selangor, Malaysia; drwadeahwafa@gmail.com; 4Paediatric Dentistry Unit, School of Dental Sciences, Universiti Sains Malaysia, Health Campus, Kota Bharu 16150, Kelantan, Malaysia; lindabaharin@usm.my; 5Department of Paediatric Dentistry and Dental Public Health, Kulliyyah of Dentistry, International Islamic University Malaysia, Kuantan Campus, Kuantan 25200, Pahang, Malaysia; drfaisal@iium.edu.my

**Keywords:** dental treatment, dentistry, primary teeth, pulpitis, pulpotomy

## Abstract

Aim: This systematic review and meta-analysis aimed to evaluate the success rates of pulpotomy treatment for irreversible pulpitis in primary teeth. Methods: This study was registered and conducted according to the Preferred Reporting Items for Systematic Reviews and Meta-Analyses Protocols. Relevant studies published between January 1980 and April 2023 were identified across eight online databases and two paediatric dentistry textbooks. Study selection, data extraction, and quality assessment were conducted by multiple investigators independently. Data analysis involved single-arm and two-arm meta-analyses, leave-one-out sensitivity analysis, meta-regression, and assessment of publication bias. The risks of bias were evaluated using the Cochrane Collaboration’s assessment tools. The levels of evidence were determined using the Oxford Centre for Evidence-Based Medicine (OCEBM) tool. Results: Five primary studies were included. The weighted mean overall success rates at 6-month and 12-month follow-ups were 97.2% and 94.4%, respectively. Two-arm meta-analysis revealed no significant difference (*p* > 0.05) between the use of mineral trioxide aggregate (MTA) and non-MTA bioceramic-based materials as pulpotomy medicaments. The sample size of each study did not affect the degree of data heterogeneity. Egger’s test revealed no significant publication bias. Conclusions: Pulpotomy may be regarded as an alternative modality for treating primary teeth with irreversible pulpitis. Nevertheless, future well-designed trials and extended follow-up periods are warranted.

## 1. Introduction

Preserving primary teeth until the eruption of permanent successors is a crucial objective in paediatric dentistry [1]. Failure to address dental caries in primary teeth can result in the involvement of the dental pulp, potentially leading to non-vital teeth [2]. Determining the pulpal status involves a combination of clinical and radiographic assessments, encompassing clinical symptoms, presence of abscess, mobility, and radiographic evaluation of furcation or periapical pathology [3]. The pulpal condition of the tooth can be classified as normal pulp, reversible pulpitis, irreversible pulpitis, and pulp necrosis [4]. However, determining whether pulpitis is reversible or irreversible is primarily based on empirical evidence.

Irreversible pulpitis is a common dental condition characterised by inflammation and infection of the dental pulp, leading to pain and discomfort. Clinically, irreversible pulpitis is diagnosed when the vital inflamed pulp is incapable of healing based on subjective and objective findings [5]. The subjective indicator of irreversible pulpitis includes persistent discomfort in response to thermal changes over an extended period, while objective findings include caries and deep restorations [4]. Moreover, the American Association of Endodontists further classifies irreversible pulpitis into symptomatic and asymptomatic subtypes. Symptomatic irreversible pulpitis is described as lingering thermal, spontaneous, or referred pain, whereas asymptomatic irreversible pulpitis exhibits no clinical symptoms, and responses to thermal testing are within normal limits with the extensive decay or fracture encroaching the pulp canal space [6].

The primary objective of treating irreversible pulpitis is to alleviate pain and prevent further infection spread while maintaining the integrity and function of the affected tooth [7]. Traditionally, the treatment of choice for irreversible pulpitis has been pulpectomy, which involves the complete removal of the infected pulp tissue or extraction [8]. However, pulpectomy has several drawbacks, such as varying success rates [9], potential expenses, time-consuming procedures, and the necessity of positive cooperation from children, which can influence treatment outcomes [10,11]. Consequently, some parents may opt for tooth extraction to avoid the stress of a prolonged procedure [12]. This may result in malocclusion and aesthetic, phonetic, and functional problems owing to the early loss of primary teeth [13].

In recent years, a paradigm shift has emerged towards performing pulpotomy, a more conservative approach that involves partial removal of the pulp tissue, in permanent teeth diagnosed with irreversible pulpitis [14]. Furthermore, a previous histological study of teeth with irreversible pulpitis showed that inflammation and microbial invasion are confined to the coronal pulp, sparing the radicular pulp [15]. This finding concurred that pulpotomy could represent a viable treatment option for teeth diagnosed with irreversible pulpitis, as it preserves tooth structure, fosters healing potential in the remaining pulp, and promotes long-term clinical and radiographic success [16]. Pulpotomy has gained popularity as an alternative treatment option to pulpectomy due to its advantages, such as reduced chair time, lower cost, and better preservation of tooth structure [1]. In primary teeth, the pulpotomy procedure is recommended in cases where caries removal leads to pulp exposure, whether the pulp is normal, exhibits reversible pulpitis, or has been traumatically exposed [17].

Despite these benefits, it is important to note that pulpotomy is not generally recommended for primary teeth diagnosed with irreversible pulpitis. This is due to concerns regarding its long-term success for primary teeth with irreversible pulpitis. Literature reports on the outcomes of primary teeth with irreversible pulpitis undergoing pulpotomy are scarce. Most studies have focused on assessing the effectiveness of pulpotomy treatment in primary teeth with carious or traumatic exposure (with normal pulp or reversible pulpitis) [1,18]. Therefore, a comprehensive evaluation of the available evidence is necessary to guide clinical decision making and establish standardised protocols for the management of irreversible pulpitis in primary teeth. This systematic review and meta-analysis aims to critically appraise the current evidence on pulpotomy treatment for primary teeth with irreversible pulpitis and to determine the overall clinical and radiographical success of this treatment approach.

## 2. Materials and Methods

### 2.1. Protocol and Registration

This systematic review and meta-analysis followed the guidelines outlined in the Preferred Reporting Items for Systematic Reviews and Meta-Analyses Protocols (PRISMA) [19]. The study was registered with the Prospective Register of Systematic Reviews (PROSPERO) at the National Institute for Health Research (NIHR), University of York, under the registration number ID: CRD42023412437.

### 2.2. Formulation of the Research Question

The research question was developed using the PICO framework. The PICO criteria include the following: (1). Problem (P): primary teeth diagnosed with irreversible pulpitis. (2). Intervention (I): pulpotomy. (3). Comparisons (C): none, pulpectomy, partial/full pulpotomy, or tooth extraction. (4). Outcome (O): clinical, radiographical, and overall success rates. Hence, the specific PICO question was formulated as follows: “What are the clinical, radiographical, and overall success rates of pulpotomy-treated primary teeth with irreversible pulpitis?” In this context, the success of pulpotomy treatment was defined as the absence of clinical and radiographic signs of failure. This included absence of spontaneous pain, tooth being non-tender to palpation or percussion, no presence of radiographic periapical radiolucency, absence of pathological root resorption (internal root resorption, external replacement root resorption, etc.), no further treatment, such as root canal therapy, being required [20]. The overall success was determined when both clinical and radiographical success was achieved.

### 2.3. Search Strategy

A comprehensive search strategy was devised to identify relevant articles published between January 1980 and April 2023. Three investigators independently performed the primary search using eight electronic databases: Google Scholar, PubMed, Web of Science, Science Direct, Cochrane Library, EBSCO, LILACS, and Open Grey. The search terms used for each database included: ‘pulpotomy’, ‘pulpotomies’, ‘coronal pulpotomy’, ‘partial pulpotomy’, ‘cvek pulpotomy’, ‘irreversible pulpitis’, ‘primary tooth’, ‘primary teeth’, ‘primary molars’, ‘primary canines’, ‘primary incisors’, ‘primary dentitions’, and ‘primary dentition’. The keywords were combined using Boolean operators ‘AND’ and ‘OR’ to construct the search strategy. In addition to the electronic database search, a manual search was conducted for two textbooks related to paediatric dentistry: “Paediatric Dentistry” [21] and “Paediatric Dentistry: A Clinical Approach” [22]. To ensure comprehensive coverage of relevant studies, the reference lists of all retrieved articles from both the electronic and manual searches were examined by one investigator using EndNote X9 (version 19.1.0.12691) software (Thomson Reuters, Stamford, CT, USA).

### 2.4. Study Selection

After removing duplicate articles using EndNote X9 software, two investigators independently screened the articles based on their titles and abstracts. Subsequently, three additional investigators conducted a comprehensive evaluation of the selected articles to identify studies that met the predefined inclusion and exclusion criteria. The inclusion criteria were as follows: (1). coronal or partial pulpotomies in primary teeth with signs and symptoms of irreversible pulpitis; (2). primary teeth with carious pulp exposure and absence pulp necrosis; (3). randomised and non-randomised clinical trials, prospective or retrospective cohort studies; (4). minimum follow-up period of 3 months after treatment; (5). clinical and radiographic findings were reported; (6). published in English language only. Meanwhile, the exclusion criteria were as follows: (1). studies involving permanent teeth; (2). traumatic pulp exposure in primary teeth; (3). other pulpal procedures: direct or indirect pulp capping, miniature pulpotomy; (4). expert opinions, commentaries, editorials, short communications, systematic reviews, literature reviews, cross-sectional studies, animal studies, case reports, and case series; (5). clinical and radiographical success not reported. Calibrations between investigators were conducted to determine the inter-rater reliability. The average concordance was calculated with the Kappa value to compare the investigators’ decisions on inclusion and exclusion [23]. Any conflicts that arose throughout the search were addressed and resolved with the assistance of another investigator.

### 2.5. Data Extraction

The study characteristics (country, year of publication, study design), patient characteristics (age, type of tooth involved, sample size), treatments (clinical and radiographic evaluation criteria, haemostasis, type of pulpotomy, type of pulpotomy medicament, and type of final restoration), and outcomes (follow-up periods, clinical and radiographical success rates) were extracted from each article using a standardised Microsoft excel spreadsheet (version 16.0) form to aid comparability. Data accuracy was verified by one investigator, and disagreements were resolved by consensus among all authors.

### 2.6. Quality Evaluation

Two distinct quality assessment methods were employed based on the study designs of the included studies. For randomised clinical trials, the Cochrane Collaboration’s tool for assessing the risk of bias in randomised trials (RoB 2) was utilised [24]. Meanwhile, non-randomised clinical trials were assessed using the Risk of Bias in Non-Randomised Studies-of Interventions (ROBINS-I) [25]. Each assessment item was meticulously assigned as either having a “high risk of bias”, “low risk of bias”, or “some concerns of bias” for RoB 2 tool (https://sites.google.com/site/riskofbiastool/welcome/rob-2-0-tool?authuser=0, accessed on 6 May 2024), and either a “low risk of bias”, “moderate risk of bias”, “serious risk of bias”, or “critical risk of bias” for ROBINS-I tool (https://www.riskofbias.info/welcome/home/current-version-of-robins-i, accessed on 6 May 2024). “NI” (No Information) was given to the assessment item if a lack of pertinent information could be found from the study. To determine the evidence level of each study, the recommendations from the Oxford Centre for Evidence-Based Medicine (OCEBM) were used [26]. To ensure a comprehensive analysis, the RoB 2 tool was reviewed independently by two investigators. Similarly, the assessment of non-randomised studies using ROBINS-I was conducted by two separate investigators. In the event of discrepancies during the quality assessment, a collaborative discussion involving the fifth and sixth investigators was initiated to reach a consensus.

### 2.7. Data Analysis

The primary outcome obtained after evaluating all the included studies was the clinical, radiographical, and overall success rates of pulpotomies in primary teeth with irreversible pulpitis. To calculate the success rate for each study, the number of successful cases was divided by the total number of cases treated at a specific follow-up assessment. The weighted mean clinical, radiographical, and overall success rates were estimated using a single-arm meta-analysis based on the DerSimonian–Laird random-effects model. This model was chosen to account for significant heterogeneity (*p* < 0.05) among the included studies at the 12-month follow-up period, as determined by the Chi-square test. The observed heterogeneity can be attributed to variations in study designs and pulpotomy medicaments [20]. Data analysis was carried out using the OpenMeta [Analyst] software (OS X 10.12 versions) with a significance level of 0.05 and 95% confidence intervals (CI). In cases where the estimated upper limit of the 95% confidence interval exceeded 1.0, the upper limit was defined as 1.0. Higgins’ *I*^2^ statistic was used to determine the degree of heterogeneity between the included studies. *I*^2^ values less than 30% indicated acceptable heterogeneity, *I*^2^ value between 30 and 60% indicated moderate heterogeneity, and *I*^2^ value greater than 60% indicated substantial heterogeneity [27]. Subgroup analysis assessing the effect of different restorative materials on the treatment success rates was not performed due to a limited number of studies available for analysis. Nevertheless, sensitivity analysis was conducted to determine the impact of each individual study on the overall results. Meta-regression analysis was conducted to assess the potential effect of sample size on the success rates. Egger’s test was used to identify publication bias.

## 3. Results

### 3.1. Study Selection

The initial search of the literature produced 1094 records from January 1980 to April 2023 (Figure 1). After removing duplicates, 474 articles were excluded. Subsequently, 567 articles were eliminated based on their titles and abstracts. The remaining articles underwent a thorough full-text assessment, adhering to the predetermined inclusion and exclusion criteria. Ultimately, five articles were deemed suitable for inclusion in this systematic review. During the study selection process, the average inter-investigator Kappa scores for the preliminary screening of titles and abstracts and the second screening involving full-text assessment were 0.79 and 0.78, respectively, indicating a ‘substantial’ level of agreement [28]. The reasons for excluding certain articles are presented in Figure 1.

The key characteristics of the included studies are summarised in Table 1. Overall, the current review comprised a total of 266 primary teeth treated with pulpotomy. Two of the studies were conducted in Iran [7,29], followed by two in Syria [30,31], and one in China [32]. Most of the primary articles were published in 2022, consisting of two randomised clinical studies [7,30], and the remaining three were non-randomised clinical studies. Among the three non-randomised studies, two were prospective cohort studies [29,31], and one was a retrospective cohort study [32]. The age range of the patients included in the current review was 3 to 9 years old, with all studies focusing on primary molars.

### 3.2. Quality Assessment of Selected Studies

For randomised clinical studies, all domains assessed using the RoB 2 tool were determined to have a “low risk of bias”. In general, three studies were classified as “low risk of bias” [7,29,30], one study was rated as “moderate risk of bias” [32], with another study rated as “high risk of bias” [31]. Additionally, two of the included studies were ranked as Level 2 evidence [7,30], while the other three studies were ranked as Level 3 evidence [29,31,32], according to the OCEBM criteria (Table 2).

### 3.3. Statistical Analysis

The clinical, radiographical, and overall success rates of pulpotomy-treated primary teeth with signs of irreversible pulpitis are shown in Table 3. Only three studies were included for the analysis of clinical and radiographical success rates at the 6-month and 12-month follow-up periods. Due to a lack of data, analysis of the success rates at 3-month, 9-month, and more-than-12-month follow-up periods was not performed. Furthermore, a study by Alawwad M et al. [31] was removed from the quantitative analysis due to a high risk of bias. The weighted mean clinical, radiographical, and overall success rates are illustrated in Figure 2 and Figure 3, for 6-month and 12-month follow-up periods, respectively. The weighted mean clinical success rates at the 6-month and 12-month follow-up periods were 97.2% (CI: (94.5, 99.9)) and 96.2% (CI: (93.0, 99.4)), respectively. The weighted mean radiographical success rates at the 6-month and 12-month follow-up periods were 97.2% (CI: (94.5, 99.9)) and 94.4% (CI: (88.7, 100)), respectively. Furthermore, the weighted mean overall success rates at the 6-month and 12-month follow-up periods were 97.2% (CI: (94.5, 99.9)) and 94.4% (CI: (88.7, 100)), respectively. The *I*^2^ of the weighted mean clinical, radiographical, and overall success rates at the 6-month follow-up period was 0%, suggesting no significant data heterogeneity. However, the *I*^2^ of the weighted mean clinical, radiographical, and overall success rates at the 12-month follow-up period ranged between 0% and 59.17%, indicating the existence of moderate data heterogeneity among the included studies.

The two-arm meta-analyses were conducted to compare different types of pulpotomy medicaments on the overall treatment success rates using odds ratios. Only two of the five included primary studies were eligible for the two-arm meta-analysis [7,30]. One study, which compared MTA with formocresol and PRF, differed from the other studies and was excluded from the analysis [31]. The 6-month and 12-month overall success rates were analysed. Pairwise analysis indicated that non-MTA bioceramic-based materials tended to demonstrate higher success rates at both 6-month (odds ratio: 0.611, CI: (0.072, 5.169)) and 12-month (odds ratio: 0.724, CI: (0.128, 4.080)) follow-up periods in treating carious primary molars with irreversible pulpitis (Figure 4). However, no significant difference was observed in the overall success rates between MTA and non-MTA bioceramic-based materials at the 6-month (*p* = 0.602) and 12-month (*p* = 0.558) follow-up periods, respectively. The *I*^2^ analysis indicated no evidence of data heterogeneity in the two-arm comparison. Nonetheless, only one study compared pulpotomy and pulpectomy [32], precluding a two-arm meta-analysis on different treatment modalities.

### 3.4. Sensitivity Analysis

Leave-one-out sensitivity analysis was performed for the overall success rates, which involved the elimination of each dataset one at a time. The highest weighted mean overall success rates at the 6-month and 12-month follow-up periods were 97.9% (CI: (94.9, 100)) and 96.9% (CI: (93.3, 100)), respectively, when Memarpour M et al. [29] was omitted. Meanwhile, the lowest weighted mean overall success rates at the 6-month and 12-month follow-up periods were 96.3% (CI: (92.3, 100)) and 91.2% (CI: (79.9, 100)) when Eshghi A et al. [7] and Alnassar I et al. [30] were excluded, respectively.

### 3.5. Meta-Regression

Meta-regression was performed to evaluate the effect of the sample size of each study on the clinical, radiographical, and overall success rates of pulpotomy-treated primary teeth with signs of irreversible pulpitis at the 6-month and 12-month follow-up periods (Table 4). No significant differences were found for all clinical, radiographical, and overall success rates at the 6-month (*p* > 0.05) and 12-month (*p* > 0.05) periods, respectively. This indicates that the degree of data heterogeneity is not directly affected by the sample size of each study. Egger’s test revealed no evidence of significant publication bias in the clinical, radiographical, and overall success rates at the 6-month (*p*-values: 0.221, 0.077, and 0.063) and 12-month (*p*-values: 0.221, 0.087, and 0.057) follow-up periods, respectively.

## 4. Discussion

The present systematic review and meta-analysis evaluated the treatment outcomes of pulpotomy in primary teeth presenting with signs and symptoms of irreversible pulpitis. The findings indicated that pulpotomy-treated teeth exhibited high clinical and radiographical success rates at the 6-month (overall success: 97.2%) and 12-month (overall success: 94.4%) follow-up periods. The high clinical and radiographical success rates of pulpotomy for primary teeth with irreversible pulpitis in the present review can be attributed to the fact that not all cariously exposed pulps were completely infected [33,34], with inflammation and microbial invasion possibly confined to the coronal pulp [15]. This advocates that primary molars with carious pulp exposure and showing signs of irreversible pulpitis can be treated with pulpotomy instead of conventional pulpectomy. Such a discovery adds another dimension to the widely held belief that pulpectomy and extraction are the appropriate treatment options for cariously exposed vital primary teeth that show evidence of irreversible pulpitis.

Irreversible pulpitis indicates that the inflammation has progressed to a point where the pulp tissue is extensively damaged and incapable of undergoing self-healing. Although a prior study found that in 84% of teeth, the clinical diagnosis of irreversible pulpitis matched the histological diagnosis [15], the precise histological finding is still unknown, and it is likely that while the coronal pulp is irreversibly inflamed, the radicular pulp may remain reversibly inflamed. Nonetheless, clinical diagnosis of irreversible pulpitis in primary teeth can sometimes be challenging, as young children might have difficulty in expressing their pain accurately [35], and the symptoms can be inconsistent. In such cases, a tooth that clinically appeared to have irreversible pulpitis might still have a portion of the pulp that can potentially heal [36]. Performing a pulpotomy allows for the removal of the affected pulp tissue while leaving a healthy portion intact, thereby increasing the chances of successful healing. In comparison to permanent teeth, primary teeth have a distinct pulpal structure [37]. The pulp in primary teeth is relatively larger in relation to the size of the tooth, and it contains more blood vessels and nerve tissue at the mid-coronal regions [38]. This increased vascularity might have contributed to better healing outcomes after pulpotomy. However, it is imperative to note that not all cases of irreversible pulpitis in primary teeth will be suitable for pulpotomy, and in some instances, such as extensive decay where the structural integrity of the tooth has been compromised, internal root resorption or presence of abscess, a more extensive treatment like pulpectomy or extraction might be necessary.

In the present review, the decline in overall success rates of pulpotomy-treated primary teeth at the 12-month follow-up is a noteworthy finding. Failure to adhere to appropriate sterilisation and disinfection protocols during the pulpotomy procedure escalates the risk of persistent bacterial presence within the tooth structure [39]. This can lead to post-treatment infections and complications. Moreover, a delay in both diagnosing and administering treatment may contribute to the diminishing success rates [40]. When a pulpotomy is executed during the advanced stages of pulp inflammation or infection, the probability of favourable outcomes decreases due to heightened damage and compromised pulp vitality. The proficiency and experience of the dental practitioner also play a pivotal role in the outcome of pulpotomy treatment [41]. Variability in skill levels among different practitioners can exert an impact on the overarching success rates. Another noteworthy aspect is the correlation between the success of pulpotomy treatment and patients’ adherence to post-treatment oral hygiene care and follow-up appointments. Neglecting recommended care practices and failing to uphold proper oral hygiene can potentially precipitate a decline in success rates. In short, proper case selection, stringent aseptic conditions, appropriate capping materials, and a well-maintained coronal seal play vital roles in ensuring the long-term success of the procedure [42]. While these factors could have collectively influenced a decline in the success rate observed during the 12-month follow-up, additional research studies are needed to offer a more comprehensive explanation.

Formocresol has long been considered the ‘gold standard’ and is commonly used in primary tooth pulpotomies. However, due to its carcinogenic properties, clinical recommendations advise against its use in paediatric dentistry. Nevertheless, a previous study conducted by Ruby et al. [43] found that 61% of certified paediatric dentists in the USA still employ it. In a recent randomised controlled trial comparing Biodentine and formocresol as pulpotomy medicaments, it has been demonstrated that both techniques exhibited comparable levels of clinical and radiographical success [44]. Although there was a trend suggesting better outcomes with non-MTA pure bioceramic-based materials, the present study did not find a statistically significant difference in overall success rates when comparing MTA to non-MTA bioceramic-based materials at both the 6-month and 12-month follow-up periods. It remains a subject of debate in the literature concerning the chemical compositions and properties of MTA and non-MTA bioceramic-based materials. MTA is predominantly composed of tricalcium silicate, dicalcium silicate, tricalcium aluminate, tetracalcium aluminoferrite, calcium sulphate, and bismuth oxide [45]. Notably, non-MTA bioceramics consisting primarily of tricalcium silicate exhibit lower levels of heavy metals, such as lead, chromium, and arsenic, compared to MTA products [46]. Furthermore, in contrast to non-MTA pure bioceramic-based cement, which includes only the calcium silicate phase, Portland cement in MTA comprises both silicate and aluminate phases. This results in various by-products of hydration (calcium silicate hydrate, calcium hydroxide, calcium aluminate hydrate, etc.) when the cement is mixed with water [47]. Previous research has shown that calcium aluminate is brittle and possesses inadequate tensile and flexural properties, necessitating reinforcement [48]. Another in vitro study showed that the mechanical properties of MTA deteriorate when it comes into contact with the tooth dentinal structure [49]. Hence, it can be hypothesised that different biomaterial compositions (MTA vs. non-MTA bioceramic-based) may impact the clinical outcomes of pulpotomy. This is crucial in the context of employing biomaterials for vital pulp therapy to ensure a more predictable outcome.

Nevertheless, the validity of the results may still be jeopardised by the possibility of errors resulting from the moderate heterogeneity of the studies included in the present systematic review. It is not possible to draw a concrete conclusion concerning the success of pulpotomy treatment in primary teeth with irreversible pulpitis based solely on clinical or radiographical evaluation. One plausible explanation is that the nerves will undergo degenerative changes, such as thickening, varicosities, and fragmentation, as primary teeth initiate the process of exfoliation and experience physiological root resorption [50]. Moreover, there is a reduction in neuronal tissue in such instances. These degenerative effects become more noticeable with increasing levels of resorption. As a result, it could be challenging to definitively label pulpotomised primary teeth without post-operative discomfort as entirely successful, as this outcome might be attributed to the contraction of neural tissue as the tooth is progressing toward its exfoliation stage. Clinical symptoms might not be adequate to determine treatment failure on their own since they might not accurately reflect the pulp’s histological condition. Patients who respond to percussion tests may have minimal or no pulpal inflammation [34]. In addition, the precise quantity of vital pulp tissue that clinicians should remove following carious pulp exposure remains a perplexing question. Profuse bleeding upon exposure suggests that either the inflamed pulp has not been entirely eliminated or that inflammation has progressed into the radicular pulp [51]. Thus, necessary adjustments to the treatment approach are warranted, potentially leading to pulpectomy if the inflammation has extended to the radicular pulp. While positive results have been observed in using pulpotomy to treat primary teeth with carious exposure, accompanied by clinical signs and symptoms of irreversible pulpitis, it is currently not feasible to establish a direct relationship due to the insufficient available data. Nonetheless, the findings offer an intriguing comparison with pulpotomies performed in permanent teeth. While pulpotomy is a well-accepted treatment for immature permanent molars, its application in mature permanent teeth with irreversible pulpitis remains a debate as a potential alternative to both extraction and conventional root canal therapy [52].

Undeniably, the success of pulpotomy in primary teeth can be influenced by several confounding factors, such as the patient’s age, cooperation level, operator skill, biological effectiveness of the pulpotomy agent, diagnostic accuracy, and quality of the final restoration [1,53]. Children under the age of three are often regarded as being in a pre-cooperative stage, in which communication cannot be established [54]. Such circumstances can indeed have a detrimental effect on the success rate of pulpotomy given the challenges posed by the child. Assessing pulp sensibility in children can be particularly challenging due to issues related to comprehension and cooperation. Nonetheless, research conducted by Hori et al. [55] concluded that the electrical pulp test is a valuable tool for determining the pulp status in primary teeth, provided there is good cooperation from the child. It is also worth noting that tooth removal is indicated when there are more than three carious teeth with likely pulpal involvement [56].

For all domains, most of the studies included in this review appeared to have a low risk of bias. One study demonstrated a moderate risk of bias for the domains ‘bias in the selection of participants into the study’ and ‘bias in the classification of interventions’ [32]. The risk and benefits of pulpotomy and pulpectomy were explained to the parents in the study, who were then obliged to choose the treatment protocol. As a result, the participants were selected for the intervention groups based on parental preference, and this might have deviated from the initial purpose of participant selection based on clinical and radiographic findings. Furthermore, a significant difference in the numbers of pulpotomy teeth (n = 88) versus pulpectomy teeth (n = 42) may have imposed the clinical and radiographic success rate of pulpotomy and led to a raised impression of pulpotomy success in primary molars with irreversible pulpitis.

In the present review, a random-effects model based on the DerSimonian–Laird method was employed to analyse the data and account for potential between-study variability [57]. This model was used as it acknowledges that the true effect size may vary across studies due to differences in methodologies and populations but also recognises that there may be common underlying characteristics that warrant inclusion in the meta-analysis to synthesise their information [58]. The data heterogeneity observed among the studies is a common challenge in meta-analyses, especially when dealing with studies of varying designs, populations, and interventions [59]. In the present review, moderate heterogeneity was observed at the 12-month follow-up period, which can be attributed to variations in study designs (randomised vs. non-randomised), haemostasis procedures (normal saline vs. sodium hypochlorite), pulpotomy medicaments (calcium-enriched mixture vs. formorcresol vs. mineral trioxide aggregate, etc.), and final restoration (stainless-steel crown vs. composite resin vs. amalgam) [20]. Meta-regression was used to explore the relationship between sample size and the observed heterogeneity among the included studies, showing that the sample size of included studies had no impact on the data heterogeneity. Nevertheless, it is crucial to interpret the results of meta-regression cautiously, as it is an exploratory analysis and may not establish causality or explain all sources of heterogeneity.

Unfortunately, the limited number of studies available for analysis prevented a meaningful subgroup analysis to assess the impact of different pulpotomy medicaments (other than MTA-based materials) and restorative materials on treatment success rates. Moreover, subgroups are sparse in terms of pulpal haemostasis during the procedure and patients’ age, causing a lack of statistical power to detect differences between the subgroups. Some included studies also did not report their subgroups findings, rendering it challenging for the present review to conduct a subgroup analysis. Nevertheless, the lack of significant publication bias in the present review suggested that the findings are not influenced by the selective publication of studies favouring positive outcomes. However, it is essential to acknowledge the possibility of publication bias due to the exclusion of non-English studies, which might have introduced language bias. To address this limitation, future systematic reviews should consider including studies published in other languages to provide a more comprehensive analysis of treatment outcomes.

The present review yielded positive outcomes for pulpotomy in primary teeth with irreversible pulpitis, but it is essential to highlight certain limitations in the study design and data interpretation. Firstly, the relatively small number of included studies limited the possibility of conducting subgroup analysis and two-arm meta-analysis for different types of pulp therapies, as well as identifying potential confounding factors that may influence treatment outcomes. The limited number of studies might be attributed to ethical concerns regarding conducting controlled trials with direct initiation of conventional pulpectomy on teeth in the control group [20]. Secondly, the presence of heterogeneity among the included studies requires cautious interpretation of the results. Observational bias may not have been completely detected in the published papers when using single-arm meta-analysis, leading to data heterogeneity [60]. Third, the follow-up periods varied across the studies, ranging from 1 week to 18 months. Longer follow-up durations could provide more valuable insights into the long-term success of pulpotomy treatment and its durability. Lastly, patient-related outcomes and patient-reported data, such as parental acceptance and patient satisfaction, should be considered in future studies to assess treatment success from a more holistic perspective. To address this, a significant amount of research should be included in the meta-analysis to ensure accurate inferential outcomes. However, it is understandable that such a requirement is rarely met, especially when addressing a new treatment modality. Despite these limitations, the present review consolidated existing data to shed light on the efficacy of pulpotomy as a viable conservative treatment option for preserving primary teeth with irreversible pulpitis. As paediatric dentistry continues to evolve, the pursuit of evidence-based practices remains critical to enhancing the oral health and well-being of children with irreversible pulpitis.

## 5. Conclusions

The present review indicates that pulpotomy may be a viable treatment option for primary teeth with irreversible pulpitis. Although the currently available data showed high clinical and radiographical success rates of pulpotomy treatment, the decline in overall success rates at 12-month follow-up raises the importance of defining treatment success more comprehensively. Despite showing positive outcomes with no significant difference between MTA and non-MTA bioceramic-based materials, it is crucial to acknowledge the moderate heterogeneity among the included studies at the 6-month follow-up and potential limitations in study design. As the field of paediatric dentistry evolves, future research should focus on conducting large-scale well-controlled randomised clinical trials with standardised protocols, longer follow-up durations, and the inclusion of patient-reported outcomes to further validate the findings of this review and guide evidence-based decision making in the management of irreversible pulpitis in primary teeth.

## Figures and Tables

**Figure 1 children-11-00574-f001:**
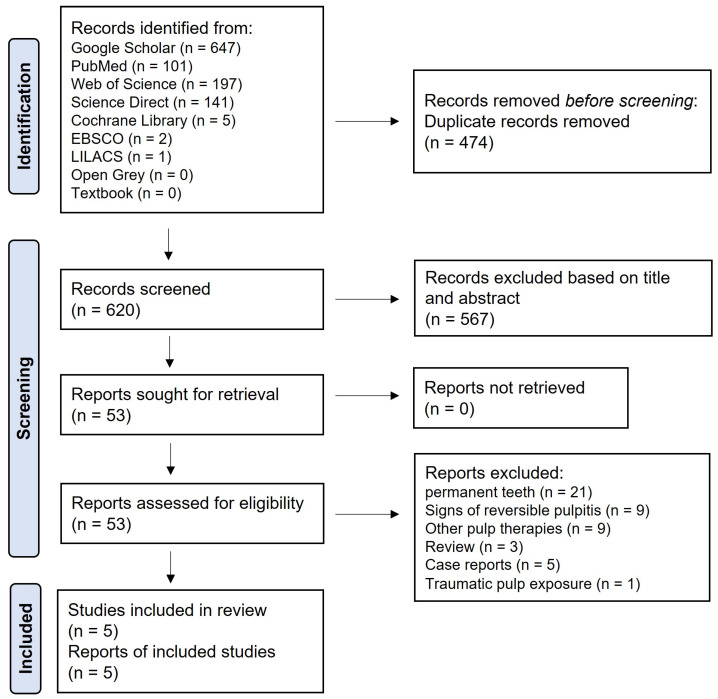
Flowchart of study selection based on PRISMA guidelines.

**Figure 2 children-11-00574-f002:**
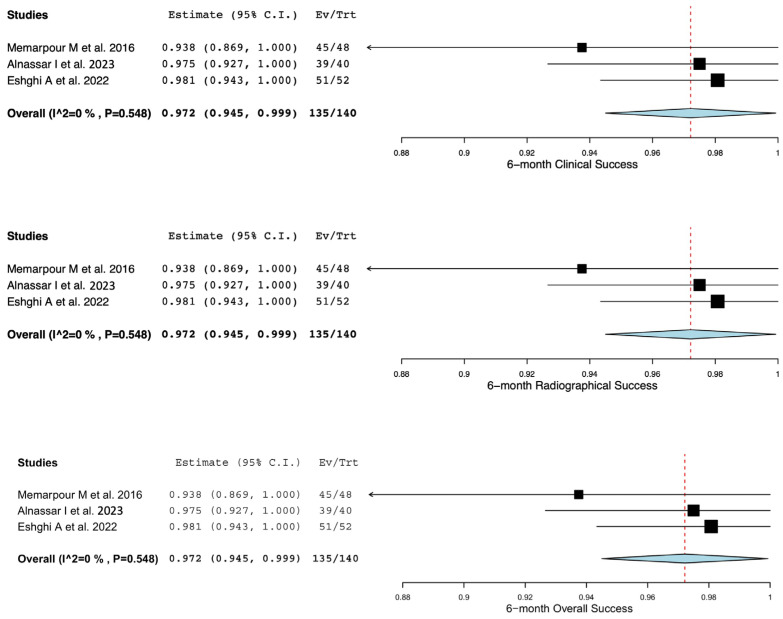
Clinical, radiographic, and overall success rates of pulpotomy-treated primary teeth with irreversible pulpitis at 6-month follow-up period [7,29,30].

**Figure 3 children-11-00574-f003:**
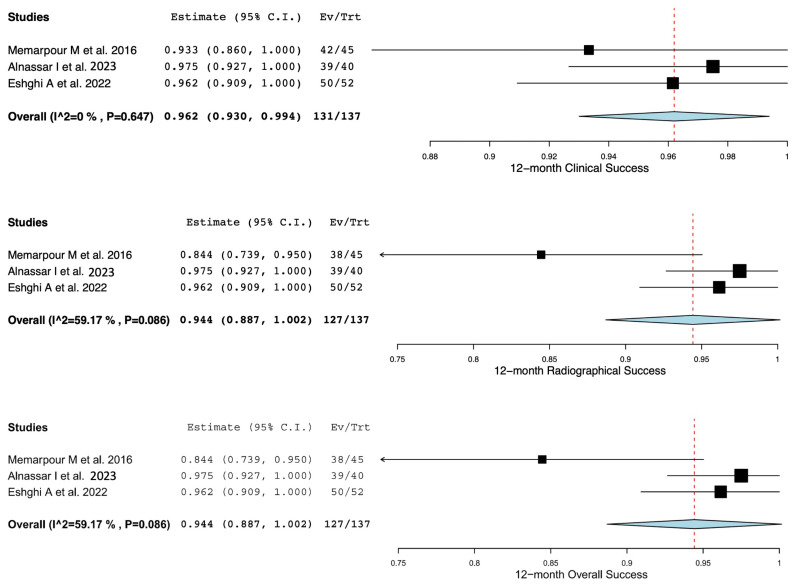
Clinical, radiographic, and overall success rates of pulpotomy-treated primary teeth with irreversible pulpitis at 12-month follow-up period [7,29,30].

**Figure 4 children-11-00574-f004:**
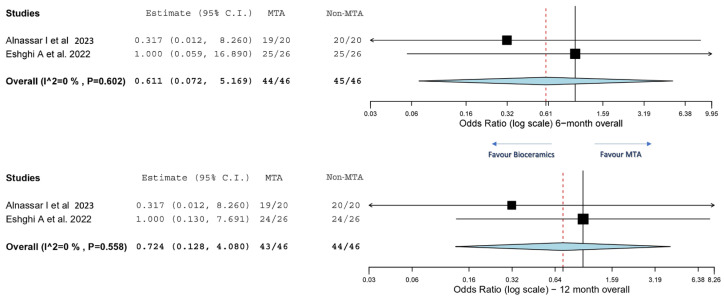
Comparison of the overall success rates of pulpotomy-treated primary teeth using MTA and non-MTA bioceramic-based material as pulpotomy medicaments at both 6-month and 12-month follow-up, respectively [7,30].

**Table 1 children-11-00574-t001:** Characteristics of the included studies.

Study (Year)	Study Design	Country	Patient Age (Years Old)	Type of Pulpotomy	Sample Size (Pulpotomy)	Teeth Involved	Haemostasis	Pulpotomy Medicaments	Final Restoration	Comparator (If Any)	Follow-Up Period	Clinical Evaluation Criteria	Radiographic Evaluation Criteria
Memarpour M et al. [29] (2016)	PC (NRCT)	Iran	6–8	Complete	50	Primary molar	NaCl—5 min	CEM	amalgam or SSC	NR	Clinical: 7 d, 3 m, 6 m, and 12 m Radiographic: 6 m and 12 m	Absence of tenderness to percussion, soft tissue redness, dental swelling, abscess, and fistula	No internal or external root resorption, no loss of lamina dura integrity, no PDL widening, and no alveolar bone resorption; but physiologically normal root resorption, radiolucency between ¼ of furcation to periapical areas and pulp canal obliteration were considered successful.
Alnassar I et al. [30] (2023)	RCT	Syria	6–8	Complete	40	Primary MD molar	2.5% NaOCl—2 min	MTA & Bioceramic putty	SSC	Grp 1—MTA Grp 2—Bioceramic putty	Clinical: 1 w, 3 m, 6 m, 9 m, and 12 m Radiographic: 1 w, 3 m, 6 m, 9 m, and 12 m	Absence of pain, swelling, fistula, pain on percussion and bites	No PDL widening, internal and external root resorption, no interradicular radiolucency, or radiolucency between 1/4 and 1/2 of furcation to periapical area
Hu X et al. [32] (2023)	RC (NRCT)	China	3–7	Complete	88	Primary molars	3% NaOCl—5 to 10 min	iRoot BP plus	CR, SSC	Vitapex pulpectomy	Clinical & radiographic: 6 m, 12 m, and 18 m	Absence of spontaneous pain, tenderness on percussion, abnormal mobility, swelling, or sinus tract.	No furcal/periapical lesion, or root resorption
Eshghi A et al. [7] (2022)	RCT	Iran	3–6	Complete	52	Primary MD second molars	NaCl—5 min	MTA & Biodentine	SSC	Grp 1—MTA Grp 2—Biodentine	Clinical: 3 m, 6 m, 12 m Radiographic: 6 m, 12 m	Absence of pain, tenderness, swelling, fistula, or pathological loosening	No root radiolucency, internal and external resorption, bone resorption, lack of integrity of the lamina dura, and PDL widening.
Alawwad M et al. [31] (2021)	PC (NRCT)	Syria	5–9	Complete	36	Primary MX and MD second molars	NaCl—5 to 15 min	Formocresol, MTA & PRF	SSC	Grp 1—Formocresol Grp 2—MTA Grp 3—PRF	6 m, 12 m	-	-

CEM: Calcium Enriched Mixture; CR: composite resin; d: day; Grp: Group; m: month; MD: mandibular; MTA: mineral trioxide aggregate; MX: maxillary; NaCl: normal saline; NaOCl: sodium hypochlorite; NR: not relevant; NRCT: non-randomised clinical trials; PC: prospective cohort; PDL: periodontal ligament fibre; PRF: platelet concentrates; RC: retrospective cohort; RCT: randomised clinical trials; SSC: stainless steel crown; w: week.

**Table 2 children-11-00574-t002:** Risk of bias and level of evidence of the included studies.

Study	The Risk of Bias in Non-Randomised Studies of Interventions (ROBINS-I)								Level of Evidence
	Bias Due to Confounding	Bias in Selection of Participants into the Study	Bias in Classification of Interventions	Bias Due to Deviations from Intended Interventions	Bias Due to Missing Data	Bias in Measurement of Outcomes	Bias in Selection of the Reported Result	Overall Risk	
Memarpour M et al. [29]	moderate	low	low	low	low	low	low	low	3
Hu X et al. [32]	low	moderate	moderate	low	low	low	low	moderate	3
Alawwad M et al. [31]	low	low	low	low	low	high	low	high	3
	**Revised Cochrane risk-of-bias tool for randomised trials (RoB 2)**								
	**Risk of bias arising from the randomization process**		**Risk of bias due to deviations from the intended interventions**	**Risk of bias due to missing outcome data**	**Risk of bias in measurement of the outcome**		**Risk of bias in selection of the reported result**	**Overall Risk**	
Alnassar I et al. [30]	low		Low	low	low		low	low	2
Eshghi A et al. [7]	low		low	low	low		low	low	2

**Table 3 children-11-00574-t003:** Clinical, radiographic, and overall success of pulpotomy-treated primary teeth with signs of irreversible pulpitis.

Author	Clinical Success					Radiographical Success					Overall Success				
	3-m	6-m	9-m	12-m	>12-m	3-m	6-m	9-m	12-m	>12-m	3-m	6-m	9-m	12-m	>12-m
Memarpour M et al. [29]	48/50	45/48	-	42/45	-	48/50	45/48	-	38/45	-	48/50	45/48	-	38/45	-
Alnassar I et al. [30]	39/40	39/40	39/40	39/40	-	39/40	39/40	39/40	39/40	-	39/40	39/40	39/40	39/40	-
Hu X et al. [32]	-	-	-	-	87/88	-	-	-	-	84/88	-	-	-	-	84/88
Eshghi A et al. [7]	-	51/52	-	50/52	-	-	51/52	-	50/52	-	-	51/52	-	50/52	-

-: No information; m: month.

**Table 4 children-11-00574-t004:** Meta-regression evaluating the effect of sample size of each study on the clinical, radiographical and overall success rates of pulpotomy on primary teeth with signs of irreversible pulpitis.

Follow-Up	Coefficient	Confidence Intervals		Standard Error	*p*-Value
		Upper Bound	Lower Bound		
**6-month**					
Clinical	0.928	1.167	0.689	0.122	0.716
Radiographical	0.928	1.167	0.689	0.122	0.716
Overall	0.928	1.167	0.689	0.122	0.716
**12-month**					
Clinical	1.008	1.280	0.737	0.138	0.737
Radiographical	0.999	1.271	0.726	0.139	0.759
Overall	0.999	1.271	0.726	0.139	0.759

## Data Availability

Not applicable.

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
