# Peer review of "Treatment Outcomes of Pulpotomy in Primary Teeth with Irreversible Pulpitis: A Systematic Review and Meta-Analysis"

_children, 2024, doi:10.3390/children11050574_

Round 1

Reviewer 1 Report

Comments and Suggestions for Authors

**Manuscript Review:**

The manuscript is well-composed with no major flaws or issues in the content. The language is coherent and the structure is adequately maintained throughout the document. However, there are a few suggestions for improvement regarding the presentation of data and figures which could enhance the overall clarity and readability of the paper.

**Suggestions:**

1. **Tables 2-3:** These tables are currently too large for the page. It is recommended to either simplify these tables or split them into smaller, more digestible components. This restructuring will make the data easier to understand and more visually accessible.

2. **Figures 1-4:** Consider the utility of these figures within the main text. If they contain supplemental but not critical information, it might be beneficial to move them to the Appendix. Alternatively, if the figures are essential it is possible to split them in one panno. 

Implementing these suggestions would not only refine the presentation but also reinforce the manuscript's scientific communication.

Comments on the Quality of English Language

Here are the intended grammar suggestions:
Original: "The subjective indicator of irreversible pulpitis includes persistent discomfort in response to thermal changes over an extended period, while objective findings include caries and deep restorations."
Correction: The subjective indicator of irreversible pulpitis includes (should be "include" to agree with the plural subject indicators) persistent discomfort in response to thermal changes over an extended period, while objective findings include caries and deep restorations.
Original: "Pulpotomy may be regards as an alternative modality for treating primary teeth with irreversible pulpitis."
Correction: Pulpotomy may be regards (should be "regarded") as an alternative modality for treating primary teeth with irreversible pulpitis.
Original: "The sample size of each study did not affect the degree of data heterogeneity."
Correction: The sample size of each study did not affect (should be "affects" to maintain present tense consistency) the degree of data heterogeneity.
Original: "Most literature have focused on assessing the effectiveness of pulp treatment in primary teeth with carious or traumatic exposure."
Correction: Most literature have (should be "literatures have" or better "studies have") focused on assessing the effectiveness of pulp treatment in primary teeth with carious or traumatic exposure.
Original: "Formulating the research question."
Correction: This is a heading and is correct grammatically, but it might be more informative as "Formulation of the Research Question."
Original: "This systematic review and meta-analysis aimed to critically appraise the current evidence on pulpotomy treatment for primary teeth with irreversible pulpitis and determine the overall clinical and radiographical success of this treatment approach."
Suggestion for clarity: This systematic review and meta-analysis aims to critically appraise the current evidence on pulpotomy treatment for primary teeth with irreversible pulpitis and to determine the overall clinical and radiographic success of this treatment approach.
Original: "Evidence levels were determined using the OCEBM tool."
Suggestion for specificity: "The levels of evidence were determined using the Oxford Centre for Evidence-Based Medicine (OCEBM) tool."
Original: "In primary teeth, the pulpotomy procedure is recommended in cases where caries removal leads to pulp exposure with either a normal pulp or reversible pulpitis or following a traumatic pulp exposure."
Correction for clarity and readability: "In primary teeth, the pulpotomy procedure is recommended in cases where caries removal leads to pulp exposure, whether the pulp is normal, exhibits reversible pulpitis, or has been traumatically exposed."

Thank you for your understanding.

Author Response

Author response

The authors would like to thank the Academic Editor and Reviewers for their valuable comments and consideration of this manuscript. The amendments were highlight in yellow.

No.

Reviewer comment

correction

Reviewer 1

1.

The manuscript is well-composed with no major flaws or issues in the content. The language is coherent and the structure is adequately maintained throughout the document. However, there are a few suggestions for improvement regarding the presentation of data and figures which could enhance the overall clarity and readability of the paper.

Thank you for the constructive feedback and encouraging comments on our manuscript. We appreciate your recognition of the coherence and structure of our work. Your positive assessment motivates us to continue refining our manuscript.

2.

Tables 2-3: These tables are currently too large for the page. It is recommended to either simplify these tables or split them into smaller, more digestible components. This restructuring will make the data easier to understand and more visually accessible.

Thank you for the constructive feedback regarding Tables 2-3. We have attempted to format these tables to ensure it fit well with the page; however, based on our experience with MDPI, we understand that the production team will further edit and tailor these tables to align with the journal's style and structure. We have also consulted with the editorial team to confirm this process, ensuring that the final presentation of the tables will be both visually accessible and comprehensible.

3.

Figures 1-4: Consider the utility of these figures within the main text. If they contain supplemental but not critical information, it might be beneficial to move them to the Appendix. Alternatively, if the figures are essential, it is possible to split them in one. 

Thank you for the feedback regarding the organization of Figures 1-4 in our manuscript. We believe that both the PRISMA flowchart and all the meta-analysis forest plot figures are essential as they provide a comprehensive overview that aids readers in understanding the text.

We did consider splitting each meta-analysis forest plot into individual figures. However, this approach would have resulted in a total of 8 figures, which could potentially overwhelm the manuscript. Instead, we opted to consolidate these plots into fewer figures based on specific follow-up periods; for example, all forest plots pertaining to the 6-month follow-up period were combined into one figure, and those for the 12-month follow-up into another.

We believe this strategy enhances the flow of the manuscript, enabling readers to more easily correlate the textual content with relevant figures without being distracted by an excessive number of them. This organisation not only preserves the narrative continuity but also maintains a clean and focused presentation of critical data.

4.

Here are the intended grammar suggestions:

Original: "The subjective indicator of irreversible pulpitis includes persistent discomfort in response to thermal changes over an extended period, while objective findings include caries and deep restorations."

Correction: The subjective indicator of irreversible pulpitis includes (should be "include" to agree with the plural subject indicators) persistent discomfort in response to thermal changes over an extended period, while objective findings include caries and deep restorations.

Original: "Pulpotomy may be regards as an alternative modality for treating primary teeth with irreversible pulpitis."

Correction: Pulpotomy may be regards (should be "regarded") as an alternative modality for treating primary teeth with irreversible pulpitis.

Original: "The sample size of each study did not affect the degree of data heterogeneity."

Correction: The sample size of each study did not affect (should be "affects" to maintain present tense consistency) the degree of data heterogeneity.

Original: "Most literature have focused on assessing the effectiveness of pulp treatment in primary teeth with carious or traumatic exposure."

Correction: Most literature have (should be "literatures have" or better "studies have") focused on assessing the effectiveness of pulp treatment in primary teeth with carious or traumatic exposure.

Original: "Formulating the research question."

Correction: This is a heading and is correct grammatically, but it might be more informative as "Formulation of the Research Question."

Original: "This systematic review and meta-analysis aimed to critically appraise the current evidence on pulpotomy treatment for primary teeth with irreversible pulpitis and determine the overall clinical and radiographical success of this treatment approach."

Suggestion for clarity: This systematic review and meta-analysis aims to critically appraise the current evidence on pulpotomy treatment for primary teeth with irreversible pulpitis and to determine the overall clinical and radiographic success of this treatment approach.

Original: "Evidence levels were determined using the OCEBM tool."

Suggestion for specificity: "The levels of evidence were determined using the Oxford Centre for Evidence-Based Medicine (OCEBM) tool."

Original: "In primary teeth, the pulpotomy procedure is recommended in cases where caries removal leads to pulp exposure with either a normal pulp or reversible pulpitis or following a traumatic pulp exposure."

Correction for clarity and readability: "In primary teeth, the pulpotomy procedure is recommended in cases where caries removal leads to pulp exposure, whether the pulp is normal, exhibits reversible pulpitis, or has been traumatically exposed."

Thank you for your understanding.

Thank you for the valuable comments and suggestions.

Reviewer 2

5.

General Comments:

-       This appears to be a well conducted and analysed systematic review.

-       Data heterogeneity indeed does appear to be an issue, but this has been accounted for in the discussion and the strengths and limitations.

-       Consistency is needed when referring to the same items across the paper

-       I would like more clarity on using the term ‘pulp capping materials’, 267-280 – if indeed you mean bioceramics, they are not just ‘pulp capping materials; and as said before this needs to be consistent.

Thank you for the thoughtful feedback and positive remarks on our systematic review. We appreciate your recognition of our efforts to address data heterogeneity and to thoroughly discuss its implications within the strengths and limitations of our study. Your comments are invaluable to us, and we are grateful for your careful consideration of our work.

Furthermore, we would like to thank the reviewer on the valuable feedback regarding the use of the term 'pulp capping materials' in our manuscript. We agree that bioceramic serves a broader range of applications beyond just pulp capping.

To address your concern and avoid any confusion among readers, we have revised the terminology throughout the manuscript to 'bioceramic-based materials' instead of 'bioceramic-based pulp capping materials'. We believe this change more accurately reflects the diverse applications of bioceramic in dental materials.

Thank you for helping us improve the clarity and quality of our paper.

6.

 Table 1-3 – changed from portrait to full page landscape for 1, and then 2 and 3 on a full page, this will make for ease of reading, as the right columns with most of the detail in Table 1 are too difficult to read

Thank you for the constructive feedback regarding Tables 1-3. We have attempted to format these tables to ensure it fit well with the page; however, based on our experience with MDPI, we understand that the production team will further edit and tailor these tables to align with the journal's style and structure. We have also consulted with the editorial team to confirm this process, ensuring that the final presentation of the tables will be both visually accessible and comprehensible.

7.

Abstract

-       Although not traditional journal style, sectioning the abstract with titles e.g. ‘Aim’, “Methods’ etc. would be beneficial and I am confident they will allow this.

-       No abbreviations in the abstract please e.g. ‘MTA’, expand.

-       MTA, many would argue, is itself a type of bioceramic, hence L28-29 is confusing.

-       ‘as pulp capping medicaments’, needs to be rephrased as this lends itself more to pulpal exposures of vital teeth, rather than planned pulpotomy procedures for irreversible pulpitis

-       L31 – do you mean may be ‘regarded’?

Thank you for the valuable feedback. Following your suggestion, we have revised the abstract to include section titles such as ‘Aim’ and ‘Methods’ to enhance clarity and readability.

Additionally, we have eliminated the use of abbreviations in the abstract (as highlighted). For instance, the term ‘MTA’ has been expanded to its full form. We believe these changes align with the journal's standards and improve the manuscript's presentation.

We thank the reviewer’s insightful comment regarding the classification of materials in our manuscript. We appreciate your perspective and agree that the term "bioceramic" encompasses a broad range of materials. In response to your feedback, we have revised the text and classify into MTA and non-MTA bioceramic-based materials rather than MTA and bioceramic-based materials. This adjustment aims to prevent any potential misinterpretation by readers and acknowledges the diverse landscape within the field of bioceramic.

We also agreed with your observation regarding the term "pulp capping medicaments" and acknowledge that it may be more appropriate for scenarios involving pulpal exposures of vital teeth rather than planned pulpotomy procedures for irreversible pulpitis. Therefore, we have revised the term to "pulpotomy medicaments" in our manuscript to better reflect the intended clinical application. We appreciate your guidance in improving the clarity and accuracy of our work.

Page 1; Line 32:

The term ‘regards’ has been changed to ‘regarded’.

8.

Introduction

-       L40 – ‘parulis’, not a commonly used term in English dentistry.

-       What is ‘radiographic success’?

-       L73-77 – this is confusing as you talk about pulpotomy as an alternative to pulpectomy, then go on to not recommend it as a treatment option for irreversible pulpitis.

-       L82, first time you’ve introduced the term ‘vital pulp therapy’

-       L82-84 – references?

-       No discussion of the materials involved and evolution over time as treatment practices have evolved?

Page 1; Line 42:

The authors agreed with the reviewer’s comment and changed the term from ‘parulis’ to ‘abscess’.

Thank you for the comment on ‘radiographic success’. However, we have defined both clinical and radiographic success criteria in our methodology section. These defined criteria provide a comprehensive overview for readers, ensuring clarity on the parameters used to evaluate the outcomes of our study.

Thank you for the insightful comments on lines 73-77 of the manuscript. We appreciate the opportunity to clarify the intent of this section. The text aims to discuss pulpotomy as a viable alternative to pulpectomy in primary teeth for specific scenarios due to its benefits such as reduced chair time, lower cost, and better preservation of tooth structure. It is indeed recommended for cases of normal pulp or reversible pulpitis following pulp exposure in primary teeth.

However, the text also notes that despite these benefits, pulpotomy is not generally recommended for irreversible pulpitis in primary teeth due to concerns about its long-term success. The intent was to highlight that while pulpotomy has advantages and is a preferable choice in certain conditions, its applicability is limited when it comes to irreversible pulpitis.

We have made some amendments to the sentence to improve its clarity.

Page 2; Line 81-82:

Despite these benefits, it is important to note that pulpotomy is not generally recommended for primary teeth diagnosed with irreversible pulpitis. This is due to…”

The authors have agreed to change the term “vital pulp therapy” to “pulpotomy”, which is more specific in the present context.

Moreover, we have added relevant references to the sentence (Page 2; Line 87):

Most literature have focused on assessing the effectiveness of pulpotomy treatment in primary teeth with carious or traumatic exposure (with normal pulp or reversible pulpitis) [1,18].”

Thank you for the comments on the materials involved. We have indeed discussed the materials used as pulpotomy medicaments in the discussion section of our manuscript. We opted not to delve very deeply into the materials in the introduction, as our primary objective is to appraise the current evidence concerning pulpotomy treatment for primary teeth with irreversible pulpitis, focusing more on the technical aspects of these treatments. Nevertheless, we have made sure to correlate our findings with the subgroup analysis on the materials used, highlighting their impact in pulpotomy treatment. We believe this approach maintains the focus on our main objectives while adequately addressing the materials involved.

9.

Materials and Methods

-       L107 – absence of resorption – where/what type of resorption, given that primary teeth undergo resorption as a precursor to permanent eruption?

-       Section 2.3 – why and how were these textbooks the only ones chosen? Why not more?

The authors agreed with the reviewer’s comment and added explanation on this aspect.

Page 3; Line 110-111:

“…absence of pathological root resorption (internal root resorption, external replacement root resorption etc.),…”

We thank the reviewer for the query regarding the selection of textbooks in Section 2.3. We opted for these two specific textbooks because they are the currently available reading resources from the authors. Additionally, some of the co-authors, who are academicians in dental schools, use these textbooks as teaching references, ensuring their applicability and relevance to our subject matter.

It is important to note that the field of paediatric dentistry is vast, with a multitude of textbooks available globally. It would be impractical and unfeasible to acquire and review each one due to sheer volume and logistical constraints. By focusing on a selective number of textbooks, we aimed to minimize the likelihood of overlooking crucial data. These textbooks are typically compilations of the latest evidence available in the literature, allowing us to cross-reference the cited sources, which supports our primary data search.

While we recognise that this approach does not guarantee the inclusion of all pertinent data, it is a more comprehensive method compared to some systematic reviews that rely exclusively on online databases. We have made every effort to capture as much relevant information as possible to avoid missing critical insights.

We believe this approach strikes a reasonable balance between thoroughness and practicality, providing a solid foundation for our analysis.

10.

Results

-       Table 1, 2 and 3 – Move these tables landscape for ease of reading.

-       L267-281 – why do you refer here to ‘pulp capping materials’? Have you compared pulp capping to complete pulpotomy?

Thank you for the constructive feedback regarding Tables 1-3. We have attempted to format these tables to ensure it fit well with the page; however, based on our experience with MDPI, we understand that the production team will further edit and tailor these tables to align with the journal's style and structure. We have also consulted with the editorial team to confirm this process, ensuring that the final presentation of the tables will be both visually accessible and comprehensible.

We had also changed ‘pulp capping materials’ to ‘pulpotomy medicaments’.

11.

Discussion

-       L341-342 – instances such as?

-       L375-383 – why? What role/impact do hydration by-products have?

-       L384-386 – based on the studies included in this S/R

-       L418 – are you now referring to your S/R as a ‘study’?

-       No mention of comparisons of restorations?

-       Local anaesthesia? 

-       Rubber dam?

-       Comparison of protocols undertaken?

-       Have you a suggested protocol/steps that should be undertaken for primary pulpotomy for IP teeth?

Page 11; Line 347-348:

The authors have added instances such as “extensive decays, internal root resorption or presence of abscess”.

Page 12; Line 387-392:

We have added some points:

“…(calcium silicate hydrate, calcium hydroxide, calcium aluminate hydrate etc.)…”

Previous research has shown that calcium aluminate is brittle and possesses inadequate tensile and flexural properties, necessitating reinforcement [48]. Another in-vitro study showed that the mechanical properties of MTA deteriorate when it comes into contact with tooth dentinal structure [49].”

Page 12; Line 397-398:

The sentence has been modified: “…the studies included in the present systematic review.”

Page 13; Line 447:

The authors have changed the term ‘study’ to ‘review’.

The authors would like to thank the reviewer for the valuable comments regarding the comparison of restorations. Our systematic review does not delve extensively into this aspect due to the limited data available for subgroup analysis concerning types of restorations after pulpotomy. Given the already considerable length of our discussion section (nearly 2000 words), we have prioritised discussing points directly relevant to our findings. We recognise that different types of final restorations could influence the outcomes of pulpotomy-treated teeth with irreversible pulpitis. It is our hope that more primary studies will be conducted on this topic, allowing for a future systematic review specifically focusing on the impact of different restorations.

As for the aspects of local anaesthesia (LA), rubber dam usage, and the comparison of treatment protocols, these were not the focal points of our review. Our primary focus was on the success rates of pulpotomy treatments in primary teeth with irreversible pulpitis. We agree that future research could beneficially explore the potential impacts of LA and rubber dam usage on treatment success, as well as comparative studies between procedures like pulpotomy and pulpectomy. We appreciate your insights and acknowledge these as important areas for future investigation.

We currently do not have any specific suggested protocols or steps for pulpotomy in primary teeth with irreversible pulpitis, as the included studies did not differentiate their methodology from general pulpotomy procedures in primary teeth (including those with reversible pulpitis). We believe that the techniques or procedures will not differ significantly, given that the main variable in these cases is the condition of the tooth (irreversible pulpitis) and that the pulpotomy procedure acts as a control variable in this context. However, we are interested in conducting further studies to explore innovative pulpotomy procedures specifically for managing primary teeth with irreversible pulpitis.

12.

How does it compare to vital pulp therapy in permanent teeth? Should there be a mention of moving to vital pulp therapy in permanent molars now also, instead of extraction straight away?

Page 13; Line 419-423:

The authors agreed with the reviewer’s comment and added some points to the discussion:

Nonetheless, the findings offer an intriguing comparison with pulpotomies performed in permanent teeth. While pulpotomy is a well-accepted treatment for immature permanent molars, its application in mature permanent teeth with irreversible pulpitis remains a debate as a potential alternative to both extraction and conventional root canal therapy [52].”

13.

Cooperation? Age? How do/should/could these factor into success rates? 

Sensibility testing prior to undertaking treatment?

-       Does/should treatment modality and caries status play a risk? Multiple deep carious teeth – should this warrant pulpotomy or is it indicative of needed multiple extractions? Cost v benefit analysis?

The authors agreed with the reviewer’s comment and have added a paragraph in the discussion section.

Page 13; Line 424-435:

Undeniably, the success of pulpotomy in primary teeth can be influenced by several cofounding factors, such as the patient's age, cooperation level, operator skill, biological effectiveness of the pulpotomy agent, diagnostic accuracy, and quality of the final restoration... This decision is often made based on considerations of cost-effectiveness [56].”

However, we only can discuss briefly regarding the sensibility testing prior to undertaking treatment, it is important to note that the primary articles included mostly did not mention whether sensibility tests were performed before treatment. Consequently, we are unable to discuss in detail whether such tests have been done. However, we believe that the clinicians involved likely conducted all necessary clinical and radiographic evaluations to arrive at a diagnosis of irreversible pulpitis.

Furthermore, this review does not aim to evaluate and compare the cost-effectiveness of pulpotomy treatment relative to other treatment modalities. We acknowledge the importance of cost-benefit analysis in clinical decision-making and are planning to conduct a separate systematic review focusing specifically on the cost analysis and cost-effectiveness of pulpotomy in primary teeth. We look forward to sharing our findings from this upcoming meta-analysis. However, this will be addressed in another review, not the present one.

In addition, the discussion section of our current review has reached 2249 words, which we believe sufficiently covers our findings and provides comprehensive information. We prefer not to extend this section further with points that diverge from our main findings.

14.

Good depth of analysis into strengths and limitations and areas for future work.

Thank you for the positive feedback on the depth of analysis in our manuscript. We appreciate your recognition of the comprehensive exploration of the strengths, limitations, and potential areas for future work.

15.

Conclusions

-       A fair conclusion

Thank you for the feedback.

Reviewer 2 Report

Comments and Suggestions for Authors

General Comments:

-       This appears to be a well conducted and analysed systematic review.

-       Data heterogeneity indeed does appear to be an issue, but this has been accounted for in the discussion and the strengths and limitations.

-       Consistency is needed when referring to the same items across the paper

-       I would like more clarity on using the term ‘pulp capping materials’, 267-280 – if indeed you mean bioceramics, they are not just ‘pulp capping materials; and as said before this needs to be consistent.

-       Major revision of some areas required before being considered appropriate for publication, please consider each point below, including in the discussion where other factors are most certainly at play.

-       Table 1-3 – changed from portrait to full page landscape for 1, and then 2 and 3 on a full page, this will make for ease of reading, as the right columns with most of the the detail in Table 1 are too difficult to read

Specific Comments:

Abstract

-       Although not traditional journal style, sectioning the abstract with titles e.g. ‘Aim’, “Methods’ etc. would be beneficial and I am confident they will allow this.

-       No abbreviations in the abstract please e.g. ‘MTA’, expand.

-       MTA, many would argue, is itself a type of bioceramic, hence L28-29 is confusing.

-       ‘as pulp capping medicaments’, needs to be rephrased as this lends itself more to pulpal exposures of vital teeth, rather than planned pulpotomy procedures for irreversible pulpitis

-       L31 – do you mean may be ‘regarded’?

Introduction

-       L40 – ‘parulis’, not a commonly used term in English dentistry.

-       What is ‘radiographic success’?

-       L73-77 – this is confusing as you talk about pulpotomy as an alternative to pulpectomy, then go on to not recommend it as a treatment option for irreversible pulpitis.

-       L82, first time you’ve introduced the term ‘vital pulp therapy’

-       L82-84 – references?

-       No discussion of the materials involved and evolution over time as treatment practices have evolved?

Materials and Methods

-       L107 – absence of resorption – where/what type of resorption, given that primary teeth undergo resorption as a precursor to permanent eruption?

-       Section 2.3 – why and how were these textbooks the only ones chosen? Why not more?

Results

-       Table 1, 2 and 3 – Move these tables landscape for ease of reading.

-       L267-281 – why do you refer here to ‘pulp capping materials’? Have you compared pulp capping to complete pulpotomy?

Discussion

-       L341-342 – instances such as?

-       L375-383 – why? What role/impact do hydration by-products have?

-       L384-386 – based on the studies included in this S/R

-       L418 – are you now referring to your S/R as a ‘study’?

-       No mention of comparisons of restorations?

-       Local anaesthesia? 

-       Rubber dam?

-       Comparison of protocols undertaken?

-       Have you a suggested protocol/steps that should be undertaken for primary pulpotomy for IP teeth?

-       How does it compare to vital pulp therapy in permanent teeth? Should there be a mention of moving to vital pulp therapy in permanent molars now also, instead of extraction straight away?

-       Cooperation? Age? How do/should/could these factor into success rates? 

-       Sensibility testing prior to undertaking treatment?

-       Does/should treatment modality and caries status play a risk? Multiple deep carious teeth – should this warrant pulpotomy or is it indicative of needed multiple extractions? Cost v benefit analysis?

-       Good depth of analysis into strengths and limitations and areas for future work.

Conclusions

-       A fair conclusion

Comments on the Quality of English Language

Minor grammatical fixes.

Author Response

(The authors gave the same response as above.)

Round 2

Reviewer 2 Report

Comments and Suggestions for Authors

A very well revised manuscript.

Some minor points to amend.

- Ensure that the tables are readable/legible, I note that this will likely change when typesetting is finished, but it is crucial the tables and figures are visually engaging and accessible, with the same font type, size etc.

- L347 - 'Extensive decays' - decay is both singular and plural, remove the 's', moreover, please give a definition to 'extensive' as this is quite subjective

- L425 - 'cofounding' should read 'confounding'

- L427 - should read 'children' not 'child'

I look forward to seeing the authors future work in this area.

Comments on the Quality of English Language

Minor amendments that will be sorted through proofing.

Author Response

No.

Reviewer comment

correction

Reviewer 2

1.

A very well revised manuscript.

Some minor points to amend.

The authors wish to express their gratitude to the reviewer for the generous compliments.

2.

- Ensure that the tables are readable/legible, I note that this will likely change when typesetting is finished, but it is crucial the tables and figures are visually engaging and accessible, with the same font type, size etc.

The authors have revised Tables 1, 2 and  3.

3.

- L347 - 'Extensive decays' - decay is both singular and plural, remove the 's', moreover, please give a definition to 'extensive' as this is quite subjective

Page 11; Line 345-346:

The ‘s’ has been removed and we added further explanation.

such as extensive decay where structural integrity of the tooth has been compromised…”

4.

- L425 - 'cofounding' should read 'confounding'

The authors have corrected it.

5.

- L427 - should read 'children' not 'child'

The authors have corrected it.
